# NovaCOMET: Open Commonsense Foundation Models with Symbolic Knowledge Distillation

**Peter West**[†‡]  **Ronan Le Bras**[‡]  **Taylor Sorensen**[†‡]
**Bill Yuchen Lin**[‡]  **Liwei Jiang**[†‡]  **Ximing Lu**[†‡]  **Khyathi Chandu**[‡]
**Jack Hessel**[‡]  **Ashutosh Baheti**[‡]  **Chandra Bhagavatula**[‡]  **Yejin Choi**[†‡]

[†]Paul G. Allen School of Computer Science & Engineering, University of Washington
[‡]Allen Institute for Artificial Intelligence
pawest@cs.washington.edu

## Abstract

We present NOVACOMET, an open commonsense knowledge model, that combines the best aspects of knowledge models and general task models. Compared to previous knowledge models, NOVACOMET allows open-format relations enabling direct application to reasoning tasks; compared to general task models like Flan-T5, NOVACOMET explicitly centers knowledge, enabling superior performance for commonsense reasoning.

NOVACOMET leverages the knowledge of opaque proprietary models to create an open knowledge pipeline. First, knowledge is symbolically distilled into NOVATOMIC, a publicly-released[1] discrete knowledge graph which can be audited, critiqued, and filtered. Next, we train NOVACOMET on NOVATOMIC by fine-tuning an open-source pretrained model. NOVACOMET uses an open-format training objective, replacing the fixed relation sets of past knowledge models, enabling arbitrary structures within the data to serve as inputs or outputs.

The resulting generation model, optionally augmented with human annotation, matches or exceeds comparable open task models like Flan-T5 on a range of commonsense generation tasks. NOVACOMET serves as a counterexample to the contemporary focus on instruction tuning only, demonstrating a distinct advantage to explicitly modeling commonsense knowledge as well.

## 1 Introduction

We present NOVACOMET, an open commonsense knowledge model combining the advantages of both knowledge models and general task models. NOVACOMET models commonsense knowledge with an open format, allowing it to be applied to general reasoning tasks in contrast to previous knowledge models. Compared to simply training

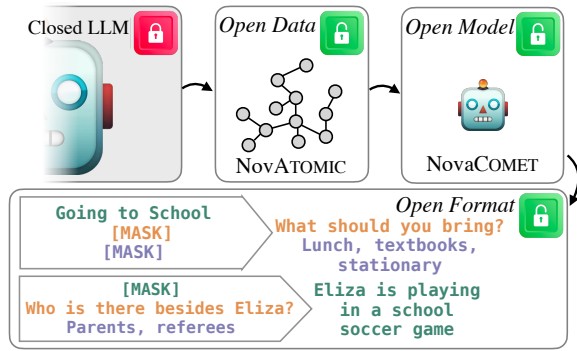

Figure 1: We leverage opaque-but-powerful proprietary LLMs into an open commonsense pipeline by: (i) creating an auditable knowledge base NOVATOMIC that gives fine-grained control over included knowledge, (ii) ensuring the generated knowledge uses a higher-coverage open-format with natural-language queries as relations and flexible mask-filling to allow for more open commonsense use-cases, (iii) demonstrating the effectiveness of (i) and (ii) via NOVACOMET's superior performance on a number of tasks, under both automatic and human evaluations.

models to be open task solvers (e.g. instruction tuning) we find that explicitly modeling knowledge in NOVACOMET also provides a distinct advantage, with NOVACOMET showing similar or superior performance to comparable open task models on a range of commonsense reasoning benchmarks.

For NOVACOMET, we leverage opaque, proprietary models like ChatGPT or GPT-4 (Ouyang et al., 2022; OpenAI, 2023) as the knowledge source in an open commonsense pipeline (Figure 1). Such models have demonstrated remarkable commonsense ability (Bubeck et al., 2023; Bian et al., 2023) yet, closed and opaque, their direct usefulness for studying commonsense is limited. Without information about training or direct access to the model, it is impossible to study *where* reported gains come from—e.g. the extent of test set contamination with benchmarks.

In our work, we use these models first to gener-

---

[1]Our resources are available at novacomet.dev

ate an **open** knowledge base (**NovAtomic**, §2.1), which can be analyzed, improved, and verified against test set contamination. Next, we train an **open** commonsense model (**NovaCOMET**, §2.3) on this knowledge: the underlying data and code will be released along with the model for the study of commonsense. This allows future testing of NovaCOMET (and of other models based on No-vAtomic) to analyze the training set—essentially allowing us to distill information from a base LLM into an auditable format.

In training NovaCOMET, we also use an **open** format: compared to previous knowledge models which use a fixed relation set and training order (head + relation→ tail) we use natural language queries as relations, and allow masked generation of all aspects of the data. This allows our model to be used in a wide range of general reasoning tasks, thus addressing a significant limitation of prior knowledge models that are limited to down-stream applications capable of effectively leveraging their restricted set of relations. Enabling an open format also allows the knowledge generation to focus on pertinent aspects of the context, rather than forcing the generation of inferences for arbitrary, potentially irrelevant relations.

Following past work on symbolic knowledge distillation (West et al., 2022), we also use No-vAtomic as the basis for training a plausibility model with human annotations (§2.2), and study how this can improve NovaCOMET (§2.3).

We test NovaCOMET on a range of commonsense generation tasks, and find that it consistently outperforms general task models of comparable size, such as Flan-T5$_{xxl}$ (Chung et al., 2022a) and T0 on commonsense tasks like abductive infilling and explanation generation. Furthermore, we assess the ability of our plausibility model to handle general commonsense QA tasks and observe that it achieves comparable or superior discriminative performance on a range of tasks. NovaCOMET will serve as an open resource for studying commonsense, and an example of the advantage of explicitly modeling commonsense knowledge in contrast to general task modeling alone.

## 2 NovaCOMET: open commonsense models

NovaCOMET is a large-scale, open commonsense model that can handle both explicit knowledge generation, and tasks that require common-

sense reasoning.

NovaCOMET is trained with symbolic knowledge distillation (West et al., 2021) by combining the commonsense data generated by large language models (§2.1) with high-quality annotations of knowledge plausibility (§2.2). We experiment with multiple methods for combining generated data with plausibility information (indicating how likely a given knowledge is) to train the final model, NovaCOMET (§2.3).

### 2.1 Generating Open Commonsense Data

Following symbolic knowledge distillation (West et al., 2021), we distill large quantities of high-quality knowledge from very large, general foundation models (§2.1.1) – we call the resulting dataset NovAtomic. One major difference from previous knowledge graphs is that we allow an open relation set, in the form of queries rather than fixed relation tokens. While commonsense knowledge often takes a *head, relation, tail* format with a fixed set of discrete relations (e.g. *X buys a lottery ticket*, **xWant**, *to win.*), we propose a **context, query, inference** (**CQI**) format with natural language queries serving as open relations. We also analyze unique properties of this distilled knowledge in §2.1.2.

### 2.1.1 Data Generation

We outline the generation process below, which consists of (1) generating contexts and (2) generating queries/inferences, resulting in our new knowledge base, NovAtomic.

**Context Generation.** First, we have experts generate 21 varied prompts to steer models to generate events or situations that require commonsense knowledge to fully understand (see B.1 for all prompts used). As variations in prompt wording influence the model's output, we use many different prompts to enhance both diversity and coverage of the generated outputs. Half of the time, we generate contexts in a zero-shot manner, while for the other half, we do one-shot generation with one example drawn from ATOMIC10X (West et al., 2022). In order to reduce generation cost, we generate 20 situations per prompt (batched generation).

We generate the contexts using GPT-3 (Brown et al., 2020) variant `text-davinci-003` (Ouyang et al., 2022) for a total cost of USD $39.56. We set `top_p=0.99` and `presence_penalty=0.3`, lowering the logit values for tokens that have already

occurred to promote diversity within each batch. Finally, to allow NovaCOMET to see some diversity of names, we also randomly swap all entities (names or "PersonX/Y/Z") for a name drawn from the 2021 public US social security application name registry[2] with probability 0.5.

**Query/Inference Generation.** As no other resource currently has examples of high-quality commonsense inferences in our proposed open format, we develop a set of few-shot examples of 10 contexts (either handwritten or selected from ATOMIC10X or from ROCStories (Mostafazadeh et al., 2016)) with 10 handwritten commonsense query/inference pairs for each (see Appendix B.2 for more details). These query/inference pairs cover a broad swathe of knowledge, including consequences, explanations, reactions, attributes, counterfactuals, etc.

For each context in NovaATOMIC generated in the previous step, we randomly select and permute $n \sim \text{Uniform}(1, 10)$ of the few-shot examples to provide in context after the instructions and then task the model with generating 10 query/inference pairs for the new context. The rationale for this random selection and permutation of the few-shot examples is to mitigate the potential overfitting to a specific ordering or selection or ordering of handwritten examples. To try to support the use case where a desired commonsense query is not known in advance, e.g. when a user simply want general knowledge for a given context, we also generated half of the commonsense hypotheses without generating a query first (prompt in B.2). At training time (§2.3), we input a `NULL` value for the query field. We generated all query/inference pairs using default decoding arguments with `gpt-3.5-turbo-0301` for a total cost of USD $337.16.

### 2.1.2 Analysis

**Comparison to Previous CSKGs.** Table 1 shows the comparisons of NovaATOMIC to existing CSKGs, ATOMIC[2020] (Hwang et al., 2020) and ATOMIC10X (West et al., 2022) in dataset statistics and lexical diversity measures. NovaATOMIC contains more diverse unique premises (heads) and hypotheses (tails) as indicated by the higher number and/or percentage of these data entries. NovaATOMIC also has higher lexical variations, as re-

[2]https://catalog.data.gov/dataset/baby-names-from-social-security-card-applications-national-data

| Type | Dataset ATOMIC | Entries # | % | 3-grams # | % |
|------|------|------|------|------|------|
| Context & Event | 2020 | 43,958 | 3.5 | 40,194 | 55.8 |
| | 10x | 165,783 | 2.6 | 235,172 | 44.9 |
| | Nova | 102,195 | 4.7 | 343,636 | 44.6 |
| Query & Relation | 2020 | 23 | - | - | - |
| | 10x | 7 | - | - | - |
| | Nova | 822,615 | 79.2 | 1,609,780 | 28.7 |
| Inference & Tail | 2020 | 602,154 | 48.3 | 847,913 | 52.5 |
| | 10x | 874,417 | 13.5 | 695,877 | 21.0 |
| | Nova | 2,030,488 | 93.2 | 5,835,099 | 30.0 |
| Total | 2020 | 1,246,582 | - | 875,157 | 51.9 |
| | 10x | 6,456,300 | - | 812,166 | 21.1 |
| | Nova | 2,178,086 | - | 7,224,608 | 28.0 |

Table 1: Statistics of NovaATOMIC compared to existing CSKG, ATOMIC[2020] and ATOMIC10X. **#** and **%** indicate the count and percentage of unique entries or 3-grams, respectively. Compared to previous CSKGs, NovaATOMIC contains more diverse entries with higher lexical variations. Notably, as NovaATOMIC adopts open data format by breaking out from fixed relation types, it contains much more diverse and flexible sets of relations denoted with questions that tie premise and hypothesis together.

flected by the significantly more diverse 3-grams. In particular, as NovaATOMIC breaks out from fixed relation types with open questions to connect premise and hypothesis, it contains much more diverse and flexible sets of relations denoted by natural language questions.

It is also of note that, based on estimates from (West et al., 2022), the total cost of ATOMIC[2020] and ATOMIC10X were approximately USD $40,000 and USD $6,000 respectively, whereas the cost for NovaATOMIC is approximately $400. Though the size of NovaATOMIC is somewhat smaller, the unit cost per example is also significantly lower.

**Analysis of Question Types.** To delve into what relations are encoded in NovaATOMIC with open questions, we conduct an analysis of question types. Figure 2 shows the top 10 most common question prefixes, including open-ended question types, such as *what* and *how*, and binary yes/no question types, such as *is* and *will*. By grouping *WH-questions* together (i.e., how, what, why, who, where, when, whose, which), we obtain 81.1% of open-ended questions and 18.9% of binary yes/no questions, indicating a diverse and flexible relation space the large portion of free-form questions represent, as shown in Figure 2(b). Table 2 shows some

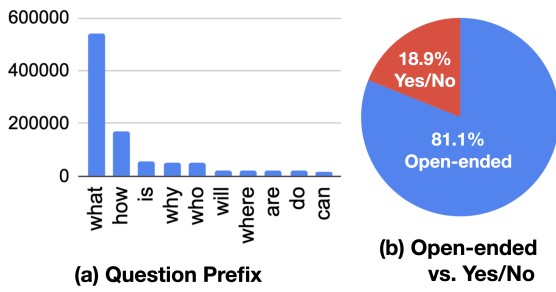

**(a) Question Prefix**  **(b) Open-ended vs. Yes/No**

Figure 2: (a) The most frequent question prefixes. (b) The composition of open-ended vs. yes/no questions.

| Most Frequent Questions |
| --- |
| What time is it? |
| Who is PersonX? |
| What is the weather like? |
| What is the prerequisite for this situation? |
| What is the consequence of the situation? |
| What is the counterfactual of the situation? |
| What will happen next? |
| What is the occasion? |
| What is the relationship between PersonX and PersonY? |
| Where are they? |

Table 2: Frequent queries in NovAtomic. Note that we take the top 100 surface forms, and cluster them into semantically related/equivalent groups by hand. Queries above represent the top groups by aggregate count, with indicative labels. See Appendix A for more details.

of the most common questions in the dataset. The most common questions are not context-specific (asking about time, weather, or location), although we find that many of the queries do condition specifically on context.

## 2.2 Plausibility Annotation

Next, we collect annotations of CQI data plausibility. Broadly, this follows West et al. (2022); Howard et al. (2023) in combining generated data with an automatic critic to maximize the quality of a final trained model. In this case, however, we explore multiple ways to incorporate annotations of plausibility into the final model NovaCOMET (§2.3.2).

Our primary means of collecting annotations of plausibility is through Amazon Mechanical Turk. We use a similar annotation template to (Hwang et al., 2020) (see Appendix C), asking annotators to decide if knowledge is *always/often*, *sometimes/likely*, *farfetched/never* true, or *invalid* (giving these annotations a score of 3, 2, 1, 0 respectively). We consider any knowledge scored 3 or 2 to be *plausible*.

In practice, we collect 20k annotations, with 1

annotator per example. For underlying data, we use 16k examples from NovAtomic, as well as 2k examples each from ATOMIC10X and Atomic$^{2020}$ to increase diversity of annotated knowledge style. While these knowledge graphs have fixed relation sets, we use sample natural language queries to replace the discrete relations (e.g. xNeed → *What did PersonX need?*).

We conduct a human agreement study on a segment of 200 examples for which we elicit 3 annotations each, finding Fleiss $\kappa$ (Fleiss, 1971) of 0.317 indicating fair agreement (Landis and Koch, 1977).

## 2.3 Training NovaCOMET

### 2.3.1 Commonsense field masking

Previous knowledge models tend to use a standard *head,relation → tail* format in training, generating some inference given the situation/concept, and one of a set of possible commonsense relations to guide generation.

The goal of NovaCOMET is maximum flexibility in handling commonsense knowledge and tasks, meaning we would like to generate any of these fields from any others. For example, we may want to generate a likely *query* that connects the *context* and *inference*; or, a context under which the query and inference are correct. To this end, we propose *commonsense field masking*, wherein we randomly sample subspans of fields to be masked for prediction, e.g.

```
Input:
 Context: Consider the list of MASK_C shows.
 Query: What is the MASK_Q show?
 Inference: Hamilton
Target:
 MASK_C = Broadway
 MASK_Q = most popular
```

The process of masking follows two steps. First, the set of which fields (CQI) will be masked is uniformly selected from all options in which at least one field is masked. Second, for each field, we randomly (with p=0.5) decide whether to mask the entire field, or a subspan. Finally, for those fields where a subspan is masked, we uniformly select the mask length, and which subspan of the given length to mask.

In effect, this gives the final model maximal flexibility at inference time. Users can mask any field, either the full span or infill a subspan as needed, allowing for use cases besides simply generating a full inference as in previous commonsense models.

We explore how this can be especially useful in §3.2.

### 2.3.2  NovaCOMET Versions

We consider a variety of methods to use the generation and critique data described above for training.

**Generation-only Model**  First, we consider the simplest option for producing a commonsense generation model: training directly on NovaAtomic. NovaCOMET$_{base}$ is trained only on generation data from §2.1 with the commonsense masking objective (§2.3.1). Plausibility is not used in this model.

**Critic-only Model**  Second, we train a standalone plausibility critic model, NovaCOMET$_{crit}$. This is trained to generate a plausibility score from a complete CQI (context, query, inference) knowledge triple, on the annotation set from §2.2. In effect, it returns a probability that a given CQI is plausible.

**Filtered Generation Model**  Following West et al. (2022), we use a simple filtering-based technique for improving generation with plausibility scores. Using NovaCOMET$_{crit}$, we calculate the probability of being plausible for all examples in NovaAtomic, and filter to only those points that meet a minimum probability. We focus on one threshold in particular, 0.99, indicating that Nova-COMET$_{crit}$ gives at least 0.99 probability to the given CQI being plausible. We call the resulting model NovaCOMET$_{filter-0.99}$, and the resulting filtered training set retains over 50% of its original size, indicating NovaAtomic is already high quality.

**Quantized Reward Conditioning**  Inspired by quantized reward conditioning in (Lu et al., 2022), we also consider more closely unifying the critical and generation data. We consider a light-weight, one-step approach (as opposed to full reinforcement learning in Lu et al. 2022) in which we annotate NovaAtomic with NovaCOMET$_{crit}$, then train a masked-prediction model that includes plausibility as a conditioning variable for predicting CQI. For annotation with NovaCOMET$_{crit}$, we greedily decode plausibility, and train a reward-conditional model NovaCOMET$_{rc}$. When decoding with NovaCOMET$_{rc}$, we condition on either of the "plausible" labels (2 or 3) from §2.2.

### 2.3.3  Model Training

We use the T5X codebase (Roberts et al., 2022) to train NovaCOMET, using the base T5 1.1 xxl (~11B parameters) checkpoint to initialize all of our experiments. We train all models on v3-128 TPU pods, using a batch size of 128 and a learning rate of 1e-5. For generation models, we train for a fixed 100k training steps, ensuring that loss does not converge or begin increasing. For models that include plausibility prediction as an objective, we stop training when evaluation loss for plausibility converges, which is often significantly before 100k training steps.

## 3  Experiments

### 3.1  Evaluating Plausibility

We begin by evaluating the performance of our plausibility model NovaCOMET$_{critic}$. Particularly, we aim to understand the ability of this model to provide a useful, absolute plausibility score. We compare the accuracy of our plausibility scores on discriminative commonsense benchmarks to judge its effectiveness.

### 3.1.1  Datasets

We consider a range of standard discriminative commonsense benchmarks: HellaSwag (**HS**) (Zellers et al., 2019) for generation recognition; $\alpha$**NLI** (Bhagavatula et al., 2019) for abductive reasoning; WinoGrande (**WG**) (Sakaguchi et al., 2019) for pronoun resolution; Commonsense QA (**CSQA**) (Talmor et al., 2019) and **CODAH** (Chen et al., 2019) for general commonsense question answering; Social IQA (**SIQA**) (Sap et al., 2019) for social commonsense; RiddleSense (**RS**) (Lin et al., 2021) for riddle answering; and Physical IQA (**PIQA**) (Bisk et al., 2019) for physical commonsense. Together, these allow us to judge the ability of models to assess the correctness/plausibility of commonsense.

### 3.1.2  Models and Baselines

As baselines, we primarily consider popular language models in a roughly similar range of parameter sizes. We include basic language model LLaMA (Touvron et al., 2023) and PaLM (Chowdhery et al., 2022) (citing performance directly for both); and language models with general task tuning such as QA for Macaw (Tafjord and Clark, 2021) or instruction tuning for Flan-T5$_{xxl}$ (Chung et al., 2022b) and T0 (Sanh et al., 2021). We create

standard-format prompts that depend on the model. When possible, models are given answer choices as input. This is an advantage over plausibility models like NOVACOMET$_{crit}$ which are designed to judge answers in isolation, but we include this to maximize baseline quality. To score options of baselines, we use negative-log-likelihood, as it was found by us to be best out of a range of options. We cite results for an alternative formatting for prompting FLAN from (Liu et al., 2023) which automatically reformats commonsense questions as statements, then judges plausibility as the likelihood of answering "yes" or "no" to whether the statement is plausible. We note that, while this method performs well, it will not generally apply to Context-Query-Inference (CQI) formatted data, as not all CQI examples can be naturally reformatted into short statements, but we include this baseline for completeness. We also cite results on GPT-3.5, ChatGPT, and GPT-4 from the same paper.

We compare baselines to NOVACOMET$_{crit}$ described in §2.3. For this models, we score options based on the probability of predicting 2 or 3 for plausibility (sometimes/likely or always/often true), renormalized against the probability of predicting 1 or 0 (rarely or never true, or invalid).

### 3.1.3 Results and Discussion

Model scores on various tasks requiring commonsense knowledge can be seen in Table 3. While various models are better at different tasks, NOVACOMET$_{crit}$ is tied for most combined 1st + 2nd place results (5). Note that the other tied system, Flan-T5 (statements) requires automatically transforming each problem into a yes or no question; a transformation that is not generally applicable to the kinds of Context-Query-Inference style problems we would like to solve when deriving commonsense information from a model.

Looking at cases where NOVACOMET$_{crit}$ fails to get the highest accuracy, it is perhaps unsurprising that PaLM 540B and 62B outperform all other models on HellaSwag, which requires predicting the most likely continuation of a scene description, a task especially well suited to a raw language model. Furthermore, with Physical IQA (PIQA), the focus is on *physical* commonsense, a subcategory that our base generator seemed to produce less naturally on inspection.

We also note that many baselines (e.g. Macaw, T0) assume access to all answer choices. For our use case (judging knowledge within NOVAATOMIC

to improve the overall dataset) we are judging examples in isolation with no clear contrastive examples. The competitive performance of NOVACOMET$_{crit}$ here, despite such disadvantages, further validates it for this use case.

### 3.2 Evaluating Generation

The central goal of NOVACOMET is in generating commonsense knowledge, and carrying out commonsense reasoning. In this section, we test the ability of various versions of NOVACOMET described in §2.3 to do this. Note that we primarily use human evaluation for model generations, following a similar setup to §2.2 with annotation templates available in Appendix C.

### 3.2.1 Datasets

First, we test the ability of models to generate commonsense knowledge in the format of previous knowledge models. Particularly, we take a sample of **ATOMIC**$^{2020}$ (Hwang et al., 2020) commonsense prompts (*head + relation*), testing the ability of models to generate a valid *tail*. Results are included in Table 4.

Next, we test on various downstream benchmarks requiring generative commonsense reasoning. First, we test abductive natural language generation ($\alpha$**NLG**) (Bhagavatula et al., 2019), wherein models must abductively fill in the gap in a story between two observations. We also consider two question-answering datasets that require commonsense reasoning: **TellMeWhy** (Lal et al., 2021) in which models explain events, and **Reflect** (Zhou et al., 2022) in which models generate ATOMIC-style inferences for dialogues. We report results for all downstream reasoning benchmarks in Table 3. We use a custom annotation template for $\alpha$NLG, and otherwise use the base CQI template from our annotation in §2.2.

### 3.2.2 Baselines and Models

For baselines, we include all of the models described in §3.1 as well as T5$_{xxl}$ ($\sim$11B parameters) finetuned for language modeling (**T5-LM**) (Raffel et al., 2019). We use straightforward prompts to describe each task and generate directly.

Different datasets can demonstrate unique ways to use the commonsense masking of NOVACOMET for generation. For example, for $\alpha$NLG, we mask between the beginning (*o1*) and ending (*o2*) events to form a natural sequence:

| system | HS | $\alpha$NLI | CODAH | WG | CSQA | SIQA | CosmosQA | RS | PIQA |
|---|---|---|---|---|---|---|---|---|---|
| **Cited Results** | | | | | | | | | |
| GPT-3.5 | 70.4 | 76.6 | 85 | 72.5 | 66.9 | 65.3 | - | - | 84.2 |
| ChatGPT | 43.0 | 60.3 | 56.8 | 61.3 | 39.6 | 52.2 | - | - | 67.6 |
| GPT-4 | 40.0 | 75.0 | 66.0 | 77.0 | 43.0 | 57.0 | - | - | 73.0 |
| Flan-T5 (statements)[1] | 64.5 | **80.8** | **89.6** | **84.7** | 69.2 | 73.2 | - | - | 83.9 |
| llama-7B[2] | 76.1 | - | - | 70.1 | - | 48.9 | - | - | 79.8 |
| llama-13B[2] | 79.2 | - | - | 73.0 | - | 50.4 | - | - | 80.1 |
| PaLM 62B[3] | 79.7 | - | - | 77.0 | - | - | - | - | 80.5 |
| PaLM 540B[3] | **83.4** | - | - | 81.1 | - | - | - | - | 82.3 |
| **Comparable General Models** | | | | | | | | | |
| Macaw | 50.8 | 71.6 | 82.9 | 60.7 | **79.4** | 68.8 | 70.4 | 58.8 | 79.4 |
| Flan-T5$_{xxl}$ | 73.5 | 70.7 | 58.7 | 72.9 | 72.8 | 55.2 | 72.9 | **60.6** | 82.0 |
| T0 | 63.7 | 70.3 | 73.4 | 58.9 | 68.1 | 66.8 | 75.4 | 53.8 | **84.9** |
| NOVACOMET$_{crit}$ | 74.4 | 80.4 | 86.7 | 79.6 | 76.7 | **77.1** | **80.3** | 58.6 | 83.4 |

Table 3: Comparison of model scores on commonsense benchmarks. **Best** results are bold and second best are underlined. Note that no other method surpasses NOVACOMET$_{crit}$ the combined number of 1st and 2nd place results (5). Comparison using *absolute scores* from different models. [1] indicates values cited from (Liu et al., 2023) which uses a pipeline with Flan-T5$_{xxl}$, [2] indicates values cited from (Chowdhery et al., 2022), [3] indicates values cited from (Touvron et al., 2023). Values for large, recent GPT models (GPT-3.5, ChatGPT, GPT4) are cited from (Liu et al., 2023).

| | $\alpha$**NLG** | | | | **Reflect** | **TellMeWhy** | ATOMIC[2020] |
|---|---|---|---|---|---|---|---|
| system | obs2 | obs1 | obs1+2 | overall | valid | valid | valid |
| **Baselines** | | | | | | | |
| LLaMA-7B | 0.030 | 0.025 | 0.022 | 0.013 | 0.388 | 0.463 | 0.470 |
| LLaMA-13B | 0.010 | 0.008 | 0.012 | 0.008 | 0.456 | 0.442 | 0.515 |
| T0 | 0.260 | 0.258 | 0.235 | 0.248 | 0.846 | 0.759 | 0.686 |
| Alpaca-7b | 0.162 | 0.123 | 0.120 | 0.122 | 0.687 | 0.852 | 0.612 |
| Alpaca-13B | 0.355 | 0.313 | 0.290 | 0.248 | 0.716 | 0.764 | 0.660 |
| Flan-Ul2 | 0.715 | 0.627 | 0.605 | 0.622 | 0.618 | 0.562 | 0.692 |
| Flan-T5$_{xxl}$ | 0.735 | 0.653 | 0.635 | 0.657 | 0.796 | 0.807 | 0.757 |
| **NOVACOMET** | | | | | | | |
| NOVACOMET$_{base}$ | 0.877 | 0.826 | 0.819 | 0.814 | 0.864 | **0.928** | 0.847 |
| NOVACOMET$_{filter-0.99}$ | **0.887** | **0.837** | **0.837** | **0.827** | 0.864 | 0.916 | 0.848 |
| NOVACOMET$_{rc(2)}$ | 0.837 | 0.793 | 0.787 | 0.797 | **0.874** | 0.916 | **0.861** |
| NOVACOMET$_{rc(3)}$ | 0.840 | 0.797 | 0.780 | 0.787 | 0.869 | 0.918 | 0.859 |

Table 4: Human evaluation of various commonsense generation tasks. Note that the basic version of NOVACOMET outperforms baselines consistently, but is outperformed by versions that use plausibility to improve. We find human agreement with Fleiss $\kappa$ (Fleiss, 1971) of 0.32, 0.44, 0.43, 0.39 (respective to order in the table) indicating fair to moderate agreement. Note, values in this table are normalized to a $[0, 1]$ range.

```
Input:
  Context: <o1> MASK_C
  Query: What happens next?
  Inference: <o2>
```

To predict a hypothesis *h* that fits between *o1* and *o2*. We found this resulted in much higher quality generations than encoding *o1, o2* as context and predicting *h* as inference.

For other datasets (Reflect, TellMeWhy, ATOMIC[2020]), we can encode examples simply by giving context and query, then predicting the

inference. For all models, we use greedy decoding.

### 3.2.3 Results and Discussion

All generation results use human evaluation, presented in Table 4. Note that human evaluation templates are included in the Appendix. We evaluate 100 examples for each system and dataset. For Reflect, TellMeWhy, and ATOMIC[2020], we use the same template as §2.2. For $\alpha$NLG we use a template measuring coherence between the generated infill and either or both hypotheses, as well as overall quality. All scores in Table 4 are normalized to

a range between 0 and 1.

Note that NOVACOMET models win across the board. Particularly effective is the filtered model NOVACOMET$_{filter-0.99}$, but so are the reward conditional models, and NOVACOMET$_{rc(2)}$ in particular, conditioned on "2" (likely/sometimes true) rather than "3" (always/often true). It is possible that answers that are always true are somewhat less creative or preferable to humans.

In general, the NOVACOMET models that use plausibility information outperform the basic NOVACOMET$_{base}$, other than on the TellMeWhy dataset. This demonstrates a particular advantage of distilling discrete data – it can be annotated, and those annotations can improve downstream performance.

Overall, superior performance of NOVACOMET suggests that explicitly modeling knowledge can provide an advantage, at least considering tasks that explicitly require commonsense knowledge and reasoning.

## 4 Related Work

**Knowledge Generation** Pretrained language models demonstrated the ability to carry implicit knowledge (Petroni et al., 2019; Dhingra et al., 2022). These large language models are prompted for generating new knowledge to perform downstream tasks such as text classification (Shin et al., 2020; Puri and Catanzaro, 2019), commonsense reasoning (Liu et al., 2022b; Trinh and Le, 2018; Davison et al., 2019). We take inspiration from commonsense LMs, designed for query commonsense knowledge, such as COMET (Bosselut et al., 2019) and COMET-2020 (Hwang et al., 2021). Domain specific LMs are also used for knowledge graph completion in specialized domains like biomedicine (Nadkarni et al., 2021). Liu et al. (2022a) use dataset cartography to prime the model with challenging examples and enable it to generate more examples with such patterns.

**Knowledge Distillation** As the process of manually creating datasets can be costly and complex, prior studies have explored the realm of automated data generation. These prior works mainly focused on extractive approaches, e.g. syntactic parsing (Zhang et al., 2020a) or pattern matching (Li et al., 2020) from unstructured text (Lehmann et al., 2015; Buck et al., 2014).

West et al. (2021) proposed filtering out low quality data using a critic model for symbolic knowl-

edge distillation from larger models. Following this, several works effectively improved upon this for iterative distillation (Sclar et al., 2022; Bhagavatula et al., 2023), self-chat with feedback and conversations with ChatGPT (Xu et al., 2023; Geng et al., 2023; Chiang et al., 2023). SODA (Kim et al., 2023) contextualized social commonsense knowledge from a knowledge graph to distill dialogues from InstructGPT. Sclar et al. (2022) established filters based on length, fidelity, and information bottleneck for distilling reference-free summarization determining the effectiveness of designing filters for selecting data for the following iteration. Recently, (Jung et al., 2023) proposed a framework to learn a high-quality model from a low-quality teacher model to distill a good dataset by summarizing and paraphrasing.

## 5 Conclusions

Overall, we introduce NOVACOMET, an open commonsense foundation model. NOVACOMET takes advantage of closed proprietary models, resulting in an open pipeline and resources that are publicly available. NOVACOMET is trained on data generated from these closed proprietary models and augmented with human annotations, resulting in both a high-quality plausibility model and improved generative model. NOVACOMET surpasses other general models of similar size at a range of commonsense knowledge-intensive tasks, demonstrating the existing need for explicit knowledge modeling, even as task-focused methods like instruction tuning grow in popularity.

## Limitations

First, we recognize that our line of research requires extensive resources and funding, limiting the broad adoption of our methodology as it is presented. Particularly, our work relies on both massive generation from proprietary language models (GPT-3 turbo) and extensive use of TPU resources. Our hope is that these barriers will only be lowered as proprietary LMs become cheaper and LMs become increasingly efficient to tune and do inference on (Dettmers et al., 2023), lowering the barrier for techniques such as ours.

Second of all, we recognize that, while we have attempted to test the query-ability of commonsense knowledge via automatic and human evaluations on a number of different tasks [FIX ME]$_{RL}$. However, current tasks are largely biased towards both

certain topics and tends to implicitly define ground truth from certain, fixed perspectives rather than acknowledging the underlying diversity of human perspectives (Santy et al., 2023). This limits our ability to assess whether our models capture genuine human agreement—or only the agreement of a certain portion of the population—something which we hope future work will investigate.

## Ethics Statement

Akin to all other machine learning approaches, our model could inadvertently exhibit biases. We acknowledge that the open format relations gathered from proprietary models may not be representative of all cultures, and thereby these perpetuate the biases that these proprietary large models possess. While generating commonsense knowledge, LLMs may result in unanticipated commonsense inferences, including those that are biased and escape our critic model. Consequently, incorporating these inferences during training can further amplify such biases. We are committed to understanding such biases and improving our critic model. However, our model's central tenet is knowledge, which contrasts with existing public models of similar size and architecture, thereby regulating the toxicity of the model. We ensured that the crowd workers involved in our project were compensated at a rate that met or exceeded the minimum wage requirement, recognizing the value of their contributions to building our model. Comparable to all open models, our model is susceptible to malicious use and it is our collective responsibility to thrust safe open usage. We acutely understand the ethical implications associated with our proposed method and are dedicated to resolving them, aiming to ensure the responsible adaptation of our approach in the community.

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

## A  Manual Cluster Analysis of Queries

To understand the contents of NOVATOMIC, we analyze the top 100 surface form queries by total count in NOVATOMIC. We cluster these queries by hand into semantically related/equivalent groups, and then further take the top 10 of these groups, displayed in the main paper text. In Table 5, we include all queries in the the top 10 clusters along with with counts and total counts per cluster.

### A.1  Automatic Evaluation of Generation

We also include automatic evaluation with 2 metrics in Table 6. We find these values show a much less distinct spread, with no model taking a clear lead over others. The seemingly lower information and general unreliability of automatic metrics was a motivation in mainly considering human evaluation.

## B  Data Generation

### B.1  Context Generation Prompts

Below are the 21 prompts used for doing context generation (delimited with "')

```
Generate 20 events.

1. Event:

'''
Generate 20 common events.

1. Event:
'''
Generate 20 everyday events.

1. Event:
'''
Generate 20 events that happen often.

1. Event:
'''
Generate 20 events that happen sometimes
    .

1. Event:
'''
Generate 20 events that include a person
    or people.

1. Event:
'''
```

```
Generate 20 everyday events about
    PersonX (one per line). It may also
    involve other entities, such as
    PersonY.

1. Event:
'''
Generate 20 situations.

1. Situation:
'''
Generate 20 common situations.

1. Situation:
'''
Generate 20 everyday situations.

1. Situation:
'''
Generate 20 situations that happen often
    .

1. Situation:
'''
Generate 20 situations that happen
    sometimes.

1. Situation:
'''
Generate 20 situations that include a
    person or people.

1. Situation:
'''
Generate 20 everyday situations about
    PersonX (one per line). It may also
    involve other entities, such as
    PersonY.

1. Situation:
'''
Generate 20 situations. They should be
    complex and include multiple parts.
    (One per line)

1. Situation:
'''
Generate 20 common situations. They
    should be complex and include
    multiple parts. (One per line)
```

| | |
|---|---|
| **What time is it?** | 800 |
| What time of day is it? | 6153 |
| What is the time of day? | 458 |
| What time of the day is it? | 328 |
| *Total* | 7739 |
| | |
| **Who is PersonX?** | 2333 |
| What is an attribute of PersonX? | 434 |
| Who are PersonX and PersonY? | 257 |
| What is PersonX? | 233 |
| Who is PersonY? | 825 |
| *Total* | 4082 |
| | |
| **What is the weather like?** | 2960 |
| What's the weather like? | 382 |
| What is the weather like outside? | 351 |
| How is the weather? | 261 |
| *Total* | 3954 |
| | |
| **What is the prerequisite for this situation?** | 640 |
| What is a prerequisite for this situation? | 517 |
| What is a prerequisite for this event? | 345 |
| *Total* | 1502 |
| | |
| **What is the consequence of the situation?** | 314 |
| What's a potential consequence of this situation? | 311 |
| What is a potential consequence of this situation? | 225 |
| What is the consequence of this situation? | 197 |
| What is a consequence of this situation? | 190 |
| What could be a consequence of this situation? | 130 |
| *Total* | 1367 |
| | |
| **What is the counterfactual of the situation?** | 570 |
| What is the counterfactual of this situation? | 202 |
| What is a counterfactual of the situation? | 163 |
| What is a counterfactual of this situation? | 125 |
| *Total* | 1060 |
| | |
| **What will happen next?** | 268 |
| What will the person do next? | 211 |
| What will PersonX do next? | 210 |
| What will they do next? | 156 |
| What might happen next? | 155 |
| *Total* | 1000 |
| | |
| **What is the occasion?** | 712 |
| What's the occasion? | 149 |
| *Total* | 861 |
| | |
| **What is the relationship between PersonX and PersonY?** | 655 |
| What is their relationship? | 198 |
| *Total* | 853 |
| | |
| **Where are they?** | 193 |
| Where is PersonX? | 233 |
| What is the setting? | 223 |
| Where is this taking place? | 135 |
| *Total* | 784 |

Table 5: The surface forms and counts included in the top 10 clusters of analyzed queries.

```
1. Situation:
'''
Generate 20 everyday situations. They
    should be complex and include
    multiple parts. (One per line)
```

```
1. Situation:
'''
Generate 20 situations that happen often
    . They should be complex and include
    multiple parts. (One per line)
```

| system | BLEU | | | | BERTScore | | | |
|---|---|---|---|---|---|---|---|---|
| | $\alpha$NLG | Reflect | TellMeWhy | ATOMIC$^{2020}$ | $\alpha$NLG | Reflect | TellMeWhy | ATOMIC$^{2020}$ |
| LLaMA-7B | 0.8 | 0.4 | 2.1 | 0.1 | 85.2 | 83.2 | 85.9 | 81.7 |
| LLaMA-13B | 1.0 | 0.5 | 4.6 | 0.1 | 85.4 | 83.8 | 86.5 | 81.7 |
| T0 | 1.3 | 3.2 | 9.0 | 0.5 | 87.2 | 88.7 | 89.1 | 85.3 |
| Alpaca-7b | 1.3 | 1.1 | 6.4 | 0.3 | 88.7 | 87.8 | 89.4 | 83.5 |
| Alpaca-13B | 2.4 | 1.2 | 6.9 | 0.2 | 88.8 | 87.4 | 89.2 | 83.3 |
| Flan-Ul2 | 4.3 | 3.4 | 5.7 | 0.5 | 90.0 | 86.5 | 85.9 | 85.3 |
| Flan-T5$_{xxl}$ | 4.3 | 4.4 | 10.8 | 0.5 | 90.0 | 88.2 | 90.2 | 86.3 |
| NOVACOMET$_{base}$ | 3.4 | 3.7 | 10.8 | 0.6 | 89.7 | 88.5 | 90.8 | 85.8 |

Table 6: Comparison of baselines and the NOVACOMET$_{base}$ using automatic scores BLEU (Papineni et al., 2002) and BERTScore (Zhang et al., 2020b). Automatic metrics do not seem to agree with human evaluation, and show less clear variation.

```
1. Situation:
'''
Generate 20 situations that happen
   sometimes. They should be complex
   and include multiple parts. (One per
   line)

1. Situation:
'''
Generate 20 situations that include a
   person or people. They should be
   complex and include multiple parts.
   (One per line)

1. Situation:
'''
Generate 20 everyday situations about
   PersonX (one per line). It may also
   involve other entities, such as
   PersonY. They should be complex and
   include multiple parts. (One per
   line)

1. Situation:
```

## B.2 Relation Generation Prompts

Below are the prompts for generating relations. To promote diversity, the number of examples were randomly selected from Uniform(1,10) and were shuffled. Some contexts come from ROCStories (Mostafazadeh et al., 2016) and (West et al., 2022), while others are handwritten. All questions and inferences are hand-written. When prompting 'gpt-3.5-turbo', we provide the instructions "Given a situation... answer" as the system message, the Context as a user message, and the ten generated queries/inferences as the system response.

## B.3 With queries

```
Given a situation, ask and answer ten
   (10) relevant questions that require
    commonsense or a world model. Some
   examples may include potential
   consequences, explanations,
   prerequisites or reactions,
   attributes, or counterfactuals. The
   commonsense facts may be about
   actors, actions, events, or ideas in
    the passage. The examples should be
    high-quality and things that are
    true. Please give a plausible answer
    at all times instead of just saying
    that it depends. Only ask questions
    that will have a relevant,
   commonsense answer.

Alisa and her family lived in Florida.
   They heard a hurricane was coming.
   They decided to evacuate to a
   relative's house. They arrived and
   learned from the news that it was a
   terrible storm.
1. What will happen now? They will wait
   out the storm.
2. How does Alisa feel? She is probably
   relieved to be out of the hurricane'
   s path.
3. What would have happened if Alisa and
    her family had not evacuated? They
   would have been in the storm.
4. Why did they decide to evacuate to a
   relative's house? They wanted to be
   in a safe place.
5. Alisa's family is what? Responsible
```

6. What might have prevented them from fleeing? If they had not heard about the hurricane, or if they had no way to get to a relative's house.
7. They would not have fled if they were not what? Cautious
8. Where does their relative live? Somewhere safe from the hurricane.
9. Should they have fled even if the storm hadn't been bad? Yes, because they might have not been able to leave if the hurricane got worse.
10. How could you describe their relative? Kind

A robber steals from a bank.
1. What are some potential characters in the situation? Robber, bank teller/worker, customers
2. Tell me something about the robber? The robber is probably armed
3. What does the robber have? The robber probably has a getaway car
4. What does the bank teller feel? The bank teller is probably scared
5. What might happen to the robber? The robber could go to jail
6. What does the bank have? The bank might have a security system
7. Before this, did the robber do anything? The robber probably planned this in advance
8. As a consequence, what will happen? After, the robber will have the money
9. What happens before this? The robber tells the bank teller to give them the money
10. How much money does the robber get? A lot of money

The woman enters the elevator
1. What did the woman have to do before? The woman had to press the button for the elevator to come to her floor
2. What is the woman's goal? The woman wants to go to a different floor
3. What will the woman do next? The woman will press the button for the floor she wants to go to

4. What could hinder this situation? The woman wants to take the stairs to be healthy
5. Is she alone? She may or may not be alone, since there could be other people in the elevator.
6. What does the woman see in the elevator? Buttons to different floors
7. What does the woman feel? The woman could feel impatient at having to wait for an elevator
8. As a consequence, what will happen? The woman will arrive at her desired floor
9. What could prevent this from happening? The elevator is out of service
10. Where are elevators located? Multi-story buildings

Emma has a big exam tomorrow. She got so stressed, she pulled an all-nighter. She went into class the next day, weary as can be. Her teacher stated that the test is postponed for next week.
1. How does Emma feel about this? Emma is probably relieved
2. Why might Emma be frustrated? Emma could be frustrated because she stayed up all night studying for nothing
3. What is the consequence of the situation? Emma will have more time to study
4. What is the prerequisite for this situation? Emma needed to have a big exam
5. Tell me what Emma will do next. Emma will probably go home and sleep.
6. What did Emma do before this? Emma was studying for her exam
7. Why did the teacher postpone the exam? The teacher may have postponed the exam because not everyone was ready.

8. What is an attribute of Emma? Emma is a procrastinator.
9. What is an attribute of Emma's teacher? flexible

10. What is the counterfactual of the situation? If Emma didn't have a big exam, she wouldn't have pulled an all-nighter.

Karen was assigned a roommate her first year of college. Her roommate asked her to go to a nearby city for a concert. Karen agreed happily. The show was absolutely exhilarating.
1. What's something we can infer about Karen? Karen likes music
2. What will happen because of this? Karen and her roommate will be better friends
3. How might this have been prevented? If Karen's roommate was shy, she might not have asked Karen to go to a concert
4. How old is Karen? Young adult
5. Why did Karen agree happily? Karen wanted to get to know her roommate better, make friends, and enjoy a concert
6. How did Karen and her roommate get to the concert? By car or public transportation
7. What's a potential consequence of this situation? Karen might have fun and meet new people
8. How does Karen feel? Karen is pleased
9. What does Karen's roommate think of her? The roommate thinks Karen is cool
10. When is the concert? The concert is likely at night

Ivette misplaced her phone at her grandparents.
1. What did Ivette do before this? Ivette was at her grandparents
2. How does Ivette feel? Ivette feels frustrated
3. What will Ivette do next? Ivette will look for her phone
4. What could hinder this situation? If Ivette was more careful
5. Ivette is what? Young
6. What would make this situation harder for Ivette? Her phone is turned off.

7. Where might her phone be? Ivette's phone could be in the house, outside, or in the car.
8. Did Ivette mean to lose her phone? No
9. What is a consequence of the situation? Ivette will have to buy a new phone
10. What would remedy the situation? Finding Ivette's phone

PersonX takes PersonY back to the hospital
1. Why did PersonX take PersonY back to the hospital? PersonY was not feeling well
2. What happened before this? PersonY was discharged from the hospital
3. What is PersonX and PersonY's relationship to eachother? They are either friends or family.
4. What would make this hard? PersonX doesn't have a car.
5. Next, what will happen? PersonY will receive medical care.
6. What happened before? PersonY asked PersonX to take them to the hospital.

7. What is PersonX? PersonX is kind
8. What is PersonY? PersonY is sick
9. Where are they? They are in a car
10. What is a result? PersonY will get better

Mila and her family lived in Florida. They heard a hurricane was coming. They decided to evacuate to a relative's house. They arrived and learned from the news that it was a terrible storm.
1. What will happen now? They will wait out the storm.
2. How does Mila feel? She is probably relieved to be out of the hurricane's path.
3. What would have happened if Mila and her family had not evacuated? They would have been in the storm.
4. Why did they decide to evacuate to a relative's house? They wanted to be in a safe place.
5. Mila's family is what? Responsible

6. What might have prevented them from fleeing? If they had not heard about the hurricane, or if they had no way to get to a relative's house.
7. They would not have fled if they were not what? Cautious
8. Where does their relative live? Somewhere safe from the hurricane.
9. Should they have fled even if the storm hadn't been bad? Yes, because they might have not been able to leave if the hurricane got worse.
10. How could you describe their relative? Kind

Alegra coyly smiled at the boy as he walked in.
1. Why did Alegra smile at the boy? Alegra was interested in him.
2. What will the boy do? The boy will notice Alegra.
3. What is Alegra's relationship to the boy? They are strangers.
4. What will happen if Alegra keeps smiling at the boy? The boy might talk to her.
5. If the boy doesn't talk to her, how will Alegra feel? Alegra will feel awkward.
6. What is the difference between a coy smile and a regular smile? A coy smile is more flirtatious.
7. How could Alegra be described? Confident
8. Where is this probably located? In a public place
9. Alegra is probably what age? A teenager or young adult
10. What would prevent this from happening? Alegra is scared to put herself out there

PersonX crosses the road
1. What is PersonX? A pedestrian
2. What could prevent this from happening? This could be prevented if there was no crosswalk.
3. What is a prerequisite for this event? A prerequisite for this event is that PersonX wants to cross the road.

4. What is something that could happen? PersonX gets hit by a car
5. If this didn't happen, what would happen? If this didn't happen, PersonX would not get to where they need to go.
6. What actors might be in this situation? PersonX, drivers, other pedestrians
7. What might PersonX be thinking? PersonX might be thinking that they need to get to the other side of the road.
8. What could be true to make PersonX reckless? PersonX crosses when there are lots of cars and no crosswalk
9. What could be true to make PersonX cautious? PersonX waits carefully for the walk signal and looks both ways.
10. What do people do before crossing the road? People might look both ways to check for cars.

Honor decides whether to bike or walk to school.
1. In what situation would Honor choose to walk to school? It is raining outside.
2. What is Honor? a student
3. How could Honor be described? Unsure
4. What would make this improbable? Honor lives very far away from the school.
5. Why might Honor choose to bike over walk? It is faster.
6. What will happen if Honor can't make up his mind? Honor will be late for school.
7. What is a possible reason for why Honor can't decide? He is feeling lazier today.
8. Either way, Honor will what? Get exercise
9. What is the difference between biking and walking? Biking is faster but requires more effort.
10. What is the weather? It might be sunny.

## B.4 Without queries

List ten (10) commonsense facts about each situation. Some examples may include potential consequences, explanations, prerequisites or reactions. The commonsense facts may be about actors, actions, events, or ideas in the passage. The outputs could also include counterfactuals or things that could hinder the event from happening. The examples should be high-quality and things that are true.

PersonX crosses the road
1. PersonX is probably going to the other side
2. Cars are on the road
3. Before this can happen, PersonX looks both ways to make sure it's safe
4. PersonX probably has a destination
5. PersonX is probably walking
6. This wouldn't happen if there wasn't a crosswalk
7. After, PersonX will be on the other side
8. If PersonX is jaywalking, they might get hit by a car
9. PersonX might use a crosswalk signal
10. This couldn't happen if the person wasn't near a road

A robber steals from a bank.
1. The robber is probably armed
2. The robber probably has a getaway car
3. The bank teller is probably scared
4. This is illegal
5. The robber could go to jail
6. The bank might have a security system
7. The robber probably planned this in advance
8. After, the robber will have the money
9. Before this happens, the robber tells the bank teller to give them the money
10. The robber might wear a mask

Addilyn and her family lived in Florida. They heard a hurricane was coming. They decided to evacuate to a relative's house. They arrived and learned from the news that it was a terrible storm.
1. They may have left valuables behind
2. They may come back to a destroyed house
3. They were smart to evacuate
4. If they didn't evacuate, they might have died
5. The hurricane was very bad
6. Now, they will wait out the storm
7. They went to their relatives house because they wanted to be in a safe place
8. They wouldn't have fled if they had not heard about the hurricane
9. The relative lives somewhere safe from the hurricane
10. Their relative is kind for letting them stay over

Fatima was assigned a roommate her first year of college. Her roommate asked her to go to a nearby city for a concert. Fatima agreed happily. The show was absolutely exhilarating.
1. Fatima has a roommate
2. Fatima likes music
3. As a result, Fatima and her roommate will be better friends
4. Fatima enjoyed the concert
5. In the future, Fatima may want to go to more concerts
6. Fatima may be more likely to spend time with her roommate
7. Fatima's roommate is considerate
8. Fatima's roommate is probably also a student
9. The roommate thinks that Fatima is cool
10. They got to the concert using a car or public transportation

Loretta misplaced her phone at her grandparents.
1. As a result, Loretta may be stressed.
2. Loretta may have to buy a new phone.
3. This event may have ruined Loretta's weekend.
4. This wouldn't happen if Loretta was more careful.
5.. Now, Loretta will probably look for her phone.

6. It will be expensive to replace her phone if it is lost.
7. Loretta is young.
8. This situation would be worse if Loretta's phone was turned off.
9. Things could be better if Loretta finds her phone
10. Loretta did not mean to lose her phone

SAN FRANCISCO - Charlotte's husband, Maxwell, was violently assaulted by a man who broke into the couple's home in San Francisco early Friday morning, the police said. The authorities identified the suspect as Lozen, 42, and said they were investigating a possible motive.
1. Lozen could be mentally ill
2. Maxwell was likely asleep when the attack happened
3. Lozen is either in custody or being searched for by the police
4. The breaking and entering was likely planned
5. Charlotte was probably not attacked
6. This would have been a frightening experience for Charlotte and Maxwell
7. If Lozen is caught, he will likely go to jail
8. Lozen's motive might have been personal
9. This wouldn't have happened if Lozen were not violent
10. Home invasions are usually premeditated

The woman enters the elevator
1. Before, the woman pushed the button for the elevator
2. The woman is going to a different floor
3. After, the woman will push the button for her floor
4. Then, she will press the button for her desired floor
5. First, the woman will wait for other people to walk out of the elevator
6. The woman might have been impatient if she had to wait for a long time
7. This couldn't happen if the elevator were out of service
8. The woman would not have done this if she wanted to take the stairs to be healthy
9. The woman may have been in a hurry
10. She is in a multi-story building

PersonX takes PersonY back to the hospital.
1. PersonY has been to the hospital before
2. The goal of PersonX was to help PersonY
3. Before this can happen, PersonY must ask PersonX to take them to the hospital
4. PersonY hopes to get medical care
5. PersonY may have been injured before this
6. This couldn't happen if PersonX does not have a car
7. PersonY is sick in some way
8. Going to the hospital may be expensive
9. This probably wouldn't happen if PersonY wasn't sick
10. PersonX cares about PersonY

Michaela has a big exam tomorrow. She got so stressed, she pulled an all-nighter. She went into class the next day, weary as can be. Her teacher stated that the test is postponed for next week.
1. Michaela is relieved that she doesn't have to take the test today
2. Michaela is sad because she worked hard to prepare and in the end didn't have to
3. When Michaela stayed up all night she was studying
4. The teacher probably postponed the exam because not everyone was ready.
5. Next week, Michaela will have to study again
6. Michaela may do better on the exam next week because she will have more time to prepare
7. If the exam was today, Michaela would have done poorly
8. The test is in a subject that

Michaela is struggling in
9. Michaela is a procrastinator
10. If Michaela hadn't stayed up all
    night, she would not be tired

Keanu decides whether to bike or walk to
    school.
1. Keanu might not choose to bike if it'
   s raining outside
2. Keanu is a student
3. They are unsure
4. If Keanu doesn't make up their mind,
   Keanu will be late for school
5. This is because Keanu is feeling lazy
    today
6. Either way, Keanu will get exercise
7. Biking is faster than walking but
   requires more effort
8. This wouldn't happen if Keanu were
   more decisive
9. This would be hard if Keanu lived
   very far away from the school
10. Keanu might choose to bike if it's a
    nice day outside

## C   MTurk Templates

Thanks for participating in this HIT!

**Evaluate the AI's guess**. Tell us, given the observation pair, how good the AI's guess is on several dimensions.

Please note that you might get the same observation pairs multiple times. For each, you will see a different AI's guess, so **please read the guess carefully**.

**IMPORTANT:**

- In this new dataset, some of the guesses may be ***exact or near copies*** of one of the observations. This is an automatic bad. Please respond with  strongly disagree  for all questions.
- Please be forgiving of minor spelling or grammar errors, as that's not what's at test.

Examples are accessible inline.

---

***Observations***

| Observation 1: | ${obs1} |
| Observation 2: | ${obs2} |

***What happened in between the observations?***

| AI's guess: | ${hyp} |

**(1)** ***Evaluate AI's guess.***

**(1.1)** AI's guess is a ***sensical*** and ***coherent*** follow-up event to Observation 1. It leaves no large unexplained information gaps.
click for examples

| strongly disagree | moderately or weakly disagree | moderately or weakly agree | strongly agree |

**(1.2)** AI's guess is a ***sensical***, ***coherent***, and ***explanatory*** preceding event to Observation 2. It leaves no large unexplained information gaps.
click for examples

| strongly disagree | moderately or weakly disagree | moderately or weakly agree | strongly agree |

**(1.3)** AI's guess is ***sensical*** and ***coherent*** when ***both Observations*** are looked at **together**. It leaves no large unexplained information gaps.
click for examples

| strongly disagree | moderately or weakly disagree | moderately or weakly agree | strongly agree |

**(2)** ***Say we were to string the sentences up as a short anecdote...***

"${obs1} ${hyp} ${obs2}"

Story ***flows well*** and is ***understandable*** (your gut judgment as a fluent English speaker).

| strongly disagree | moderately or weakly disagree | moderately or weakly agree | strongly agree |

(Optional) Please let us know if anything was unclear, if you experienced any issues, or if you have any other fedback for us.

Submit

Figure 3: MTurk template for $\alpha$ NLG.

(WARNING: This HIT may contain adult content. Worker discretion is advised.)

Thanks for participating in this HIT!

**You will evaluate how often assertions are true. Each assertion is comprised of 3 parts:** *Phrase A*, *Question*, *Phrase B*

| *Phrase A*, *Phrase B* | Short phrases. May describe objects, object properties, events, actions, etc. |
| *Question* | How *A* relates to *B*. |

**For each assertion, determine how true it is:**

| *always/often* | Always or quite often true. |
| *sometimes/likely* | Sometimes is true or true for some people. -or- Likely true. |
| *farfetched/never* | False or farfetched, at best. -or- Unlikely to be true. |
| *invalid* | This assertion makes no sense (i.e., "what does this even mean?!"). |
| *too unfamiliar to judge* | Cannot make a fair evaluation. Unfamiliar with one or both of the phrase. |

**If you see "nothing in particular" for** *Phrase B*, assess Phrase B in context:

- Sometimes certain actions can simply be responded to by doing nothing!
- Other times, doing nothing in particular is simply a weird or unlikely reaction to something.

**New!** Please report any **prejudiced or inappropriate language**:

- Profane or offensive content (NSFW, R-rated material etc)
- Prejudiced assumptions or derogatory language that villainizes people.
  HOWEVER, please note, not all negative content is derogatory especially if Phrase B is intrinsically what Phrase A means. For example:
  *criminals* how are they characterized? *committing crime* is **OK**.
  ↳ This isn't necessarily villianizing people since "criminal" means "a person who has commited a crime".
  *homeless* how are they characterized? *being lazy* is **prejudiced**.
  ↳ There are many reason a person is rendered homeless. This is a gratuitous prejudice about homelessness.
- Material that people may find disturbing, off-putting, or improper

A couple NOTES:

- Please be **forgiving** of *spelling or grammatical errors*
- If the terms are too obscure or you don't know the truth of the fact at the top of your head, it is okay to mark is "too unfamiliar to judge". If you can answer (e.g., based on likelihood), please provide a response.

Examples (click to expand/collapse)

1) **${context1}**
$\{query1\}$
*${inference1}*

***How often does the assertion hold true?***

| always/often | sometimes/likely | farfetched/never | invalid | too unfamiliar to judge |

☐ This fact is true but outdated
☐ I would count this as an inappropriate, prejudiced or offensive material

2) **${context2}**
$\{query2\}$
*${inference2}*

***How often does the assertion hold true?***

| always/often | sometimes/likely | farfetched/never | invalid | too unfamiliar to judge |

☐ This fact is true but outdated
☐ I would count this as an inappropriate, prejudiced or offensive material

3) **${context3}**
$\{query3\}$
*${inference3}*

***How often does the assertion hold true?***

| always/often | sometimes/likely | farfetched/never | invalid | too unfamiliar to judge |

☐ This fact is true but outdated
☐ I would count this as an inappropriate, prejudiced or offensive material

4) **${context4}**
$\{query4\}$
*${inference4}*

***How often does the assertion hold true?***

| always/often | sometimes/likely | farfetched/never | invalid | too unfamiliar to judge |

☐ This fact is true but outdated
☐ I would count this as an inappropriate, prejudiced or offensive material

5) **${context5}**
$\{query5\}$
*${inference5}*

***How often does the assertion hold true?***

| always/often | sometimes/likely | farfetched/never | invalid | too unfamiliar to judge |

☐ This fact is true but outdated
☐ I would count this as an inappropriate, prejudiced or offensive material

(Optional) Please let us know if anything was unclear, if you
experienced any issues, or if you have any other fedback for us.

Submit

Figure 4: MTUrk template for CQI.