# OpenReview forum: "NovaCOMET: Open Commonsense Foundation Models with Symbolic Knowledge Distillation"
_EMNLP/2023/Conference — EMNLP 2023 Findings_

### Official Review · Reviewer_a9Hp · 2023-07-23

**Typos Grammar Style And Presentation Improvements:** See reasons to reject above.
**Soundness:** 3

**Excitement:**

2: Mediocre: This paper makes marginal contributions (vs non-contemporaneous work), so I would rather not see it in the conference.

**Missing References:**

These papers are also relevant to this work:

Wang, J., Qu, J., Liang, Y., Li, Z., Liu, A., Liu, G., & Zheng, X. (2023). Snowman: A Million-scale Chinese Commonsense Knowledge Graph Distilled from Foundation Model. arXiv preprint arXiv:2306.10241.

Bhagavatula, C., Hwang, J. D., Downey, D., Bras, R. L., Lu, X., Sakaguchi, K., ... & Choi, Y. (2022). I2d2: Inductive knowledge distillation with neurologic and self-imitation. arXiv preprint arXiv:2212.09246.

Wang, C., Li, J., Chen, Y., Liu, K., & Zhao, J. (2022, December). CN-AutoMIC: Distilling Chinese Commonsense Knowledge from Pretrained Language Models. In Proceedings of the 2022 Conference on Empirical Methods in Natural Language Processing (pp. 9253-9265).

Kim, H., Hessel, J., Jiang, L., Lu, X., Yu, Y., Zhou, P., ... & Choi, Y. (2022). Soda: Million-scale dialogue distillation with social commonsense contextualization. arXiv preprint arXiv:2212.10465.

**Paper Topic And Main Contributions:**

This paper introduces NovaCOMET, an innovative open commonsense foundation model that combines the benefits of knowledge models and general task models. The authors employ a robust pipeline to extract knowledge from expensive, closed large language models and implement stringent quality controls to create an open commonsense knowledge base called NovATOMIC. Subsequently, they use NovATOMIC to train NovaCOMET, an open commonsense foundation model, using a masked commonsense modeling objective. Empirical findings demonstrate that NovaCOMET outperforms other models in both commonsense question-answering and generation tasks.

The key contributions of this paper are the creation of two new resources: NovATOMIC, an open commonsense knowledge base, and NovaCOMET, a pre-trained commonsense foundation model.

**Questions For The Authors:**

My questions and suggestions are listed above in reasons to reject. I am open to hearing the authors' rebuttal to my concerns and would be willing to reconsider my evaluation if necessary.

**Reasons To Accept:**

The paper presents two valuable contributions, NovATOMIC and NovaCOMET. NovATOMIC is a commonsense knowledge base that is constructed by distilling LLM with an open relation set and strict quality critic filtering. This approach achieves both diversity and high quality, resulting in a rich and expansive knowledge base. NovaCOMET is a generative model trained upon NovATOMIC that is both high-performing and easy to deploy. These resources will be highly valuable for future research.

Furthermore, the authors' innovative approach to designing their prompting pipeline is noteworthy. For instance, they introduce the context, query, and inference (CQI) paradigm for commonsense knowledge formulation, as well as the commonsense field masking technique, which enhances the generative model's robustness during training.

The empirical results demonstrate NovaCOMET's high performance on various tasks, confirming the quality of its data source NovATOMIC.

**Reasons To Reject:**

While the paper presents solid work, I feel that it lacks novelty. It appears to me that the authors have simply reproduced the symbolic knowledge distillation (ATOMIC10X) pipeline with a more advanced LLM (davinci003). Aside from the resulting resources, the paper's novel contributions are twofold: the construction of an open relation set and masked commonsense generation.

Regarding the first contribution, while the open relation set removes a traditional constraint of commonsense knowledge bases, it also raises the difficulty of using NovATOMIC. There will no longer be systematic relations, only queries, and contexts. This loss of formalization, in exchange for diversity and flexibility, may make it challenging to structure and raise more challenges to apply to downstream tasks. I’m also thinking about the feasibility of manually expanding the relations in ATOMIC10X to accommodate the prompts in NovATOMIC, they may achieve the same objective.

Masked commonsense modeling seems to be applying T5's baseline training style, which is not a novel paradigm, just a different method for training a generative model. It is a countermeasure to deal with the loss of relations in NovATOMIC.

For the experiments, it would be beneficial to include comparisons with GPT3.5, ChatGPT, and even GPT4 if the authors have sufficient funding for access. This will make the paper more outstanding, even if NovaCOMET fails to outperform them. Additionally, filling in the missing blanks in Cited results in Table 3 is necessary, as the current comparisons seem awkward, with most results left blank.

The paper's structure and writing could also be improved. While following the writing pipeline of ATOMIC10X is acceptable, it can also be messy. Consider separating NovATOMIC and NovaCOMET into two sections and adding briefings at the start of each main section to inform readers what they will be reading. Some titles, such as NovaCOMET Versions, could be improved by changing them to training strategies. It is also better to state NovATOMIC in the title of respective sections.

**Reproducibility:**

4: Could mostly reproduce the results, but there may be some variation because of sample variance or minor variations in their interpretation of the protocol or method.

**Reviewer Confidence:**

4: Quite sure. I tried to check the important points carefully. It's unlikely, though conceivable, that I missed something that should affect my ratings.

---

> ### Author Rebuttal · Authors · 2023-08-29
>
> We greatly appreciate the reviewer’s careful attention to our work, finding our contribution “highly valuable for future research” as well as “high-performing and easy to deploy”. We also thank the reviewer for their stated willingness to reconsider their evaluation.
>
> We believe that reviewer concerns can be fully addressed here and in the final version. In particular, we will report expanded baseline results as requested by the reviewer, with a large portion included in the table below, and the full set to be included in the final paper (we are currently time-limited by GPU resources and OpenAI quota).
>
> **“missing blanks in Cited results in Table 3 is necessary, as the current comparisons seem awkward, with most results left blank.”**
>
> We include a sample of these new results in the table below (for Llama models on a subset of tasks). Full results are currently running but will require another week to complete. These will be included in the final paper version.
>
> **“include comparisons with GPT3.5, ChatGPT, and even GPT4”**
> Thank you for this suggestion! We will include these comparisons in the final paper. We include some discriminative comparisons in the table below, cited from previous work, but will run full discriminative and generative evaluation for these 3 models in the final version, once our OpenAI quota/budget allows this.
>
>
> **“authors have simply reproduced the symbolic knowledge distillation (ATOMIC10X) pipeline with a more advanced LLM (davinci003)”**
> As the reviewer points out, our work uses Symbolic Knowledge Distillation to allow an open relation set (new in commonsense models) while ATOMIC10X only used existing relations. As well:
>
> - Unlike ATOMIC10X, our critic model is applicable to discriminative commonsense tasks in general as it is not constrained by closed relations (see also response to Reviewer eW5y for a combined generator-critic model)
>
> - Resulting NovaCOMET can be applied to general-format commonsense generation tasks, unlike past commonsense work in this vein
>
> - Our masking strategy results in diverse use-cases, both filling in full fields (context, query, inference), or parts of multiple fields.
>
>
> **“seems to be applying T5's baseline training style”**
> We would like to clarify that our objective differs from the T5 baseline style, specifically in explicitly targeting use-cases of our context-query-inference data format. It allows for masking of anywhere from single words to a full field (context, query, inference), and explicitly accounts for field-structure by not masking across boundaries. In contrast, the T5 objective simply samples random subspans to infill with a set distribution, not accounting for generation use-cases or structure of the underlying text.
>
> **“loss of formalization, in exchange for diversity and flexibility, may make it challenging to structure and raise more challenges to apply to downstream tasks”**
> While we agree that the data/model no longer operate in a fully structured space, they can still be applied to the existing static relation set of ATOMIC. This is clear from the ATOMIC column in Table 4: NovaCOMET is still very effective at previous ATOMIC relations, while the flexibility allows application to extended downstream tasks (e.g. aNLG).
>
> NovATOMIC is indeed less structured than previous versions of ATOMIC. We plan to continue expanding NovATOMIC through generation, and argue that as it becomes more dense, NovATOMIC will contain larger structured subgraphs following the previous ATOMIC relations, as these are a strict subset of our open-format queries and denote natural and common queries.
>
> **“manually expanding the relations in ATOMIC10X to accommodate the prompts in NovATOMIC, they may achieve the same objective.”**
> We find that the queries (effective relations) in NovATOMIC are quite varied (800k unique queries, from Table 1 in the paper) and so it may be difficult to capture all of these with a static set of relations. However, we agree that expanded relations would narrow this gap, and as we note in our response above, some subset of NovATOMIC likely corresponds (with varying degrees of strictness) to existing relations. We will include more analysis on the semantic variety of queries in the final paper.
>
>
> **Improvements to structure and new citations**
> We thank the reviewer for the suggestions! We agree that the changes in format will greatly increase readability of the paper, and will incorporate suggested improvements into the final version. We will also include all suggested citations.
>
> **Tables**
>
>
> | model     |   HS |   aNLI |   CODAH |   WG |   CSQA |   SIQA |   PIQA |
> |:----------|-----:|-------:|--------:|-----:|-------:|-------:|-------:|
> | GPT-3.5   | 70.4 |   76.6 |    85   | 72.5 |   66.9 |   65.3 |   84.2 |
> | chatGPT   | 43   |   60.3 |    56.8 | 61.3 |   39.6 |   52.2 |   67.6 |
> | GPT-4     | 40   |   75   |    66   | 77   |   43   |   57   |   73   |
> |:--
> | Llama-7B  | 76.1 |   62.7 |    40.6 | 70.1 |   43.8 |   48.9 |   79.8 |
> | Llama-13B | 79.2 |   62.9 |    42.5 | 73   |   41.2 |   50.4 |   80.1 |
> |:--
> | NC (us)   | 74.4 |   80.4 |    86.7 | 79.6 |   76.7 |   77.1 |   83.4 |
> Table: Sample of extended results to be added. GPT-3.5, chatGPT, and GPT-4 results are currently cited from Liu et al. 2023 but we will run these ourselves once OpenAI quota allows, along with full generative evaluation. Llama values include new experimental results to fill in Table 3 as suggested, as well as those cited from Chowdhery et al. 2022. We will fill in all missing columns for cited models once GPU resources allow this.

---

### Official Review · Reviewer_4pDd · 2023-07-28

**Soundness:** 4

**Excitement:**

4: Strong: This paper deepens the understanding of some phenomenon or lowers the barriers to an existing research direction.

**Paper Topic And Main Contributions:**

This paper proposes to use an LLM (GPT3.5) to generate synthetic data for commonsense reasoning. The proposed dataset, NOVATOMIC, is designed in the format of context, query, inference, which overcomes the limitations of fixed relation sets in previous models. The authors demonstrate that their model, NOVACOMET, which is a finetuned T5 on the NOVATOMIC data with a commonsense field masking objective, outperforms strong vanilla LM baselines on multiple relevant datasets.
Contributions: 1. A new synthetic dataset NOVATOMIC for common sense reasoning. The quality of the data seems good, with ample plausibility annotation from crowdsourcing, and the authors promised to open-source the data. The design of the data in the (context, query, inference) format is interesting and supports open-format relations.
2. Training a plausibility model to filter the data is a good idea.
3. The authors showed that the data generation pipeline and the finetuning strategy are successful to distill commonsense knowledge from LLM.

**Questions For The Authors:**

1. It is good to see the data analysis in Section 2.1.2 and the distribution of the type of questions the data covers. Can you show if the open-form questions are enough to cover the relations in previous datasets/models like COMET?

2. Section 2.3.1 introduces an interesting masking strategy that focuses on entities. What is the benefit of this compared to simpler masking strategies or next-word prediction tasks?

**Reasons To Accept:**

1. The contributions are clear and significant.
2. The paper provides a detailed analysis of the data distribution, which adds to the credibility of the proposed dataset.
3. The experiments are thoroughly described and the models are evaluated on multiple relevant datasets and compared to reasonable baselines.
4. The writing is clear and easy to understand.

**Reasons To Reject:**

1. Despite that the writing is very clear, some evaluation steps are not fully replicable. For example, in the evaluation of the generation model, only human annotation scores are provided on a subset of 100 examples from each dataset, which may limit the reproducibility of the results. It would be good to include some automated scores on the entire evaluation sets.
2. The authors did not discuss the limitations of representing commonsense knowledge in the newly proposed context, query, inference format.
3. The baselines the authors have compared their approach to are all general LMs that were not specifically trained for commonsense reasoning tasks.

**Reproducibility:**

3: Could reproduce the results with some difficulty. The settings of parameters are underspecified or subjectively determined; the training/evaluation data are not widely available.

**Reviewer Confidence:**

4: Quite sure. I tried to check the important points carefully. It's unlikely, though conceivable, that I missed something that should affect my ratings.

**Typos Grammar Style And Presentation Improvements:**

Line 60, the period before "Next".

Line 438 and line 450, I think you mean Table 4

---

> ### Author Rebuttal · Authors · 2023-08-29
>
> We thank the reviewer for an extensive review, including finding our “contributions are clear and significant”.
> We believe all concerns can be fully addressed in this rebuttal, and by updating the paper. We particularly note that **we include requested automatic evaluation scores** and will add this evaluation to the final version.
>
>
> **“only human annotation scores are provided on a subset of 100 examples” … “include some automated scores on the entire evaluation sets”**
>
> We are glad to follow this suggestion! We have **added automatic evaluation,** and **include tables of sample results below** for 2 metrics: BLEU (using SacreBLEU) and BERTScore. Broadly, NovaCOMET is on par with or exceeding baselines, as in the human evaluation.
>
> We will include a more extensive set of metrics in the paper, e.g. BLEU, ROUGE, METEOR, BERTScore, BLEUERT, and MoverScore.
> We will also extend human evaluation to 200 examples.
>
> **“did not discuss the limitations of representing commonsense knowledge in the newly proposed context, query, inference format”**
>
> We agree with the importance of these limitations, and plan to incorporate a full section on this using the extra page in the camera ready. For example, while context-query-inference format allows more flexible application, it also allows more misuses. Constrained relation sets of previous commonsense models made exploitation/misuse difficult by forcing only certain generation functions, while open-format NovaCOMET does not have this intrinsic safeguard.
>
>
> **“compared their approach to … general LMs that were not specifically trained for commonsense reasoning tasks”**
>
> While we agree that general LMs are not trained for commonsense tasks specifically, we argue that NovaCOMET is not explicitly trained for most commonsense tasks either. Rather, it is tuned to focus on commonsense knowledge, but has not been trained on e.g. aNLG data before testing on it. Further, general LMs have become some of the strongest models for commonsense tasks, and so serve as a natural comparison point.
>
> We will add commonsense models (e.g. COMET-distil, COMET-2020) where applicable as further baselines, although early experiments suggest significantly weaker performance than general LM baselines.
>
>
> **“Can you show if the open-form questions are enough to cover the relations in previous datasets/models like COMET?”**
>
> One piece of evidence in favor of this is the “ATOMIC” column in table 4 of the paper, which tests the ability of models to generate on COMET/ATOMIC data. NovaCOMET models exceed baselines, and we are happy to add previous COMET models as further baselines for this experiment. This demonstrates a strong ability of open-format queries to cover previous relations.
>
>
> **“What is the benefit of this compared to simpler masking strategies or next-word prediction tasks?”**
>
> Our masking strategy is specifically designed for generation use-cases, allowing partial or full masking of any of the 3 text fields (context, query, inference). In comparison, simpler masking strategies (e.g. basic T5 masking) would not account for field structure, or allow for the same variance in generation length as our strategy, which can mask anything from 1 word to a full field.
>
> Next-word prediction tends to offer strong generative performance, but without the flexible use-cases of our technique, for instance generating the most likely query (middle field) while holding the context and inference constant.
>
>
> **Typos**
>
> Thank you for pointing out typos! We will fix these in the final paper version.
>
> **Tables**
>
> | model      |   anlg |   reflect |   tmw |   atomic |
> |:-----------|-------:|----------:|------:|---------:|
> | T0         |    1.3 |       3.2 |   9   |      0.5 |
> | Flan-T5    |    4.3 |       4.4 |  10.8 |      0.5 |
> | Flan-UL2   |    4.3 |       3.4 |   5.7 |      0.5 |
> | LLaMA-7B   |    0.8 |       0.4 |   2.1 |      0.1 |
> | LLaMA-13B  |    1   |       0.5 |   4.6 |      0.1 |
> | Alpaca-7B  |    1.3 |       1.1 |   6.4 |      0.3 |
> | Alpaca-13B |    2.4 |       1.2 |   6.9 |      0.2 |
> | NovaCOMET  |    3.4 |       3.7 |  10.8 |      0.6 |
> **Table A:** Automatic evaluation with SacreBLEU score (huggingface). NovaCOMET achieves the top score on 2 of 4 datasets (TellMeWhy and ATOMIC-2020) and second highest on Reflect.
>
> | model      |   anlg |   reflect |   tmw |   atomic |
> |:-----------|-------:|----------:|------:|---------:|
> | T0         |   87.2 |      88.7 |  89.1 |     85.3 |
> | Flan-T5    |   90   |      88.2 |  90.2 |     86.3 |
> | Flan-UL2   |   90   |      86.5 |  85.9 |     85.3 |
> | LLaMA-7B   |   85.2 |      83.2 |  85.9 |     81.7 |
> | LLaMA-13B  |   85.4 |      83.8 |  86.5 |     81.7 |
> | Alpaca-7B  |   88.7 |      87.8 |  89.4 |     83.5 |
> | Alpaca-13B |   88.8 |      87.4 |  89.2 |     83.3 |
> | NovaCOMET  |   89.7 |      88.5 |  90.8 |     85.8 |
> **Table B:** Automatic evaluation using BERTScore. Similar to human evaluation, NovaCOMET seems to be on par with, or superior to most baseline methods, including achieving the top score on the TellMeWhy dataset, and second-highest score on Reflect and ATOMIC-2020. BertScores are significantly more clustered than human evaluation scores.

---

### Official Review · Reviewer_eW5y · 2023-08-04

**Typos Grammar Style And Presentation Improvements:** N/A
**Soundness:** 2

**Excitement:**

3: Ambivalent: It has merits (e.g., it reports state-of-the-art results, the idea is nice), but there are key weaknesses (e.g., it describes incremental work), and it can significantly benefit from another round of revision. However, I won't object to accepting it if my co-reviewers champion it.

**Missing References:**

N/A

**Paper Topic And Main Contributions:**

This paper introduces NovaCOMET, a new commonsense knowledge model that allows open format relations and enables direct application to reasoning tasks. The authors first collect NovaATOMIC by symbolic distillation with GPT3 and open format relations. Then the authors fine-tunes a T5 model on the generated commonsense knowledge data using a open format objective. The authors pre-train both generative and discriminative models. Results show that the discriminative model achieves competitive results compared to other comparable models; and the generative model archives better results on commonsense generation tasks compared to other comparable general models.

**Questions For The Authors:**

N/A

**Reasons To Accept:**

1. The paper is in general well written and easy to follow.
2. The idea of training a open format commonsense knowledge model is well motivated and can be very helpful for some applications.

**Reasons To Reject:**

1. The technical novelty is limited because the dataset and model are constructed and trained with existing methods. This is not a big issue though because the main contribution is the collection and release of the dataset and the model.
2. While the authors claim that NovaCOMET and solve general tasks while also serve and an open format commonsense knowledge model, the experiments on discriminative tasks are using the critic model while it is the generation model that be used as a commonsense knowledge model. The authors also claim the model can serve as a counterexample for instruction tuning models. However, the experiments are only done on commonsense tasks and need two (generation and critic) models. This does not supports the claim.
3. One of the main contribution of NovaCOMET is the support of open format. However, the authors do not conduct much analysis and evaluation on this part. This severely limits the completeness of the work.


**Reproducibility:**

4: Could mostly reproduce the results, but there may be some variation because of sample variance or minor variations in their interpretation of the protocol or method.

**Reviewer Confidence:**

4: Quite sure. I tried to check the important points carefully. It's unlikely, though conceivable, that I missed something that should affect my ratings.

---

> ### Author Rebuttal · Authors · 2023-08-29
>
> We thank the reviewer for finding our work “well motivated and very helpful for some applications” and our paper “well written and easy to follow”
> We address all concerns below, especially highlighting **new experiments to combine the critic+generation model** into a single model, thus handling a primary concern of the reviewer, that two models are needed. We also carried out a **topic analysis** of open-format queries to be added to the final paper.
>
> **“[experiments] need two (generation and critic) models”, “discriminative tasks … [use] the critic model”**
>
> To address this concern, that two models are needed for the given experiments (critic + generator), we have trained a **new model** combining both the generation and critic objectives with multi-task training. We **include initial results below** (Tables A and B), showing that combining the models into one general commonsense model **does not significantly affect performance.** Thus, we can release a single **general commonsense NovaCOMET model** with the capabilities of both original models.
>
>
> **“main contribution of NovaCOMET is the support of open format …  authors do not conduct much analysis and evaluation on this part”**
>
> We agree that further analysis will enrich the paper, and will include more insights about the open format of NovaCOMET and NovATOMIC in the added page for the final version. For instance, we have carried out a **topic analysis of NovATOMIC** on the open-format queries using BERTopic. We find novel query topics that extend beyond ATOMIC, e.g. asking about the time of the given context (example topic query: “what time of day is it?”). We also find topics that cover existing ATOMIC relations, e.g. the xReact relation (example topic query: “what emotions is the person feeling”).
>
> We will include further details in the final paper version. This analysis uses 10K examples from NovATOMIC, but this will also be expanded.
>
> In terms of evaluating the open format of the model NovaCOMET, we note that open format is a key aspect of generative evaluation: neither the TellMeWhy nor aNLG tasks fit existing ATOMIC relations, so this evaluation explicitly demonstrates the flexibility of the open-format approach compared to previous commonsense models.
>
>
> **“dataset and model are constructed and trained with existing methods”**
>
> We agree with the reviewer’s observation that one “main contribution is the collection and release of the dataset and the model”, and also note that this is a new style of open commonsense model, and thus analysis/testing of such a model is also a proposed contribution. This style of symbolic distillation is a relatively new method, and understanding it requires extensive examples and resources which make use of it.
>
> **“counterexample for instruction tuning models. However, the experiments are only done on commonsense tasks”**
>
> We clarify that the focus of our work is specifically solving commonsense tasks, and NovaCOMET should be seen as a counterexample to instruction tuning in this domain. Instruction models have been among the most effective options for such tasks recently, which is why they are our main point of comparison. We will clarify our explicit focus on the commonsense domain in the final version of the paper.
>
>
>
> **Tables**
>
> | model    |   aNLI |   CODAH |   Cosmos |   CSQA |   CSQA2 |   HS |   PIQA |   RS |   SIQA |   WG |
> |:---------|-------:|--------:|---------:|-------:|--------:|-----:|-------:|-----:|-------:|-----:|
> | critic   |   0.8  |    0.87 |     0.89 |   0.77 |     0.5 | 0.74 |   0.83 | 0.59 |   0.77 | 0.8  |
> | combined |   0.82 |    0.86 |     0.89 |   0.76 |     0.5 | 0.74 |   0.84 | 0.56 |   0.78 | 0.78 |
> **Table A:** Comparing performance of the new, combined NovaCOMET model to our existing NovaCOMET critic model on discriminative tasks. The combined model does not show a significant decrease in performance, with very similar average accuracies (0.784 vs 0.781)
>
> | model    |   aNLG o2 |   aNLG o1 |   aNLG o1+o2 |   aNLG all |   Reflect |   TMW |   ATOMIC |
> |:---------|:----------|:----------|:-------------|:-----------|:----------|:------|:---------|
> | gen-only |      0.74 |      0.82 |         0.72 |       0.72 |      0.87 |  0.86 |     0.86 |
> | combined |      0.73 |      0.82 |         0.71 |       0.72 |      0.86 |  0.84 |     0.84 |
> **Table B:** Comparing the existing generation-focused NovaCOMET model to the new, combined NovaCOMET model on generation tasks. The combined model is able to achieve very similar generation results by human eval to the generation-only model.

---

### Meta-Review · Area_Chair_47rq · 2023-09-18

**Recommendation:** 3

**Metareview:**

Reviewers almost unanimously agree that the purpose of the paper is quite clear. The work is quite well-motivated. Some analyses were missing in the manuscript, such as comparison with GPT-4, ChatGPT, etc, but were later added to the rebuttal. However, everyone also unanimously agree that, while the produce of the paper is important and useful, the core methodology is nothing new--except open-format prompting--, which hurts the overall scientific contribution.

# Pros
1. NovaComet significantly outperforms the baseline models (which include many recent proprietary LLMs) on benchmark commonsense QA tasks by significant margin.
2. The open-format masking strategy is interesting and has lead to rich synthetic data, as verified by some reviewers.

# Cons (and suggestions)
1. Reviewer eW5y points out that the paper lacks analysis on the open-format prompting, despite it being its main scientific contribution. The authors should report on the coverage of ATOMIC relations, as promised.
2. The authors should report the combined results with the critic model.
3. The core methodology of generating synthetic data from LLM and finetuning another model is not new at this point.

---

### Decision · Program_Chairs · 2023-10-07

**Decision:**

Accept-Findings

**Comment:**

Reviewers almost unanimously agree that the purpose of the paper is quite clear. The work is quite well-motivated. Some analyses were missing in the manuscript, such as comparison with GPT-4, ChatGPT, etc, but were later added to the rebuttal. However, everyone also unanimously agree that, while the produce of the paper is important and useful, the core methodology is nothing new--except open-format prompting--, which hurts the overall scientific contribution.

# Pros
1. NovaComet significantly outperforms the baseline models (which include many recent proprietary LLMs) on benchmark commonsense QA tasks by significant margin.
2. The open-format masking strategy is interesting and has lead to rich synthetic data, as verified by some reviewers.

# Cons (and suggestions)
1. Reviewer eW5y points out that the paper lacks analysis on the open-format prompting, despite it being its main scientific contribution. The authors should report on the coverage of ATOMIC relations, as promised.
2. The authors should report the combined results with the critic model.
3. The core methodology of generating synthetic data from LLM and finetuning another model is not new at this point.